# Long-Acting Opioid Analgesics for Acute Pain: Pharmacokinetic Evidence Reviewed

Betty M. Tyler  and Michael Guarnieri *

Department of Neurosurgery, Johns Hopkins University, 1550 Orleans Street, 1550 Orleans St. CRB II, Baltimore, MD 21287, USA; btyler@jhmi.edu
* Correspondence: mguarnie@jhmi.edu; Tel.: +1-410-614-1791

**Simple Summary:** Pharmacokinetic (PK) studies measure the time for a dose of a drug to reach therapeutic levels in a patient's blood or tissues. This data provides a simple and precise method to compare the response to drugs made by different manufactures or the same drug provided in different delivery formats; for example, oral tablets, intravenous infusions, or subcutaneous (SC) injections. The present study provides the first review of PK data for buprenorphine, an opioid analgesic, in animals dosed with the drug in a lipid or polymer carrier. The PK data from studies using the polymer carrier system illustrate greater variability, an outcome probably related to the different manufacturing conditions of the supplying vendors. The lipid product was produced by a single vendor.

**Abstract:** Long-acting injectable (LAI) opioid formulations mitigate the harm profiles and management challenges associated with providing effective analgesia for animals. A single dose of a long-acting opioid analgesic can provide up to 72 h of clinically relevant pain management. Yet, few of these new drugs have been translated to products for veterinary clinics. Regulatory pathways allow accelerated drug approvals for generic and biosimilar drugs. These pathways depend on rigorous evidence for drug safety and pharmacokinetic evidence demonstrating bioequivalence between the new and the legacy drug. This report reviews the animal PK data associated with lipid and polymer-bound buprenorphine LAI formulations. Buprenorphine is a widely used veterinary opioid analgesic. Because of its safety profile and regulatory status, buprenorphine is more accessible than morphine, methadone, and fentanyl. This review of PK studies coupled with the well-established safety profile of buprenorphine suggests that the accelerated approval pathways may be available for this new family of LAI veterinary pharmaceuticals.

**Keywords:** analgesia; buprenorphine; harm reduction; long acting; pharmacokinetic; extended release



## 1. Introduction

Opioids are frontline drugs for animals and humans to manage the pain from trauma and surgery. As a class of drugs, opioids are well-tolerated, fast acting, and effective. Decades of research and clinical experience provide a broad understanding of their metabolism, interactions with other drugs, and confidence in their safety when used as prescribed. Contrary to experience with nonsteroidal anti-inflammatory agents and locally acting anesthetics, evidence of organ damage is rare. Nonetheless, there are clinical impressions of suboptimal opioid use for pain management in veterinary medicine [1,2]. Buprenorphine, morphine, and fentanyl have 4–6 h half-lives in vivo. Parenteral or oral doses at 6–8 h intervals are prescribed for up to 3 days to assure clinically significant pain management. However, the short half-life of opioid analgesics deters their use in animals [3,4]. A major concern is that repeated restraint of animals for oral dosing or injections with pain medication stresses the animal and increases the opportunities for unintended injuries.

Veterinary drug investigators have studied LAI analgesics to address this challenge. Based on the lower toxicity, increased compliance, and overall decreased costs of medical

care, extended-release formulations are widely used in human clinical settings. Grant and colleagues confirmed the strategy for laboratory animals by formulating morphine with a phospholipid drug carrier [5]. A single subcutaneous (SC) drug injection provided 2–3 days of neuropathic pain therapy. Subsequent studies confirmed the approach using polymer-bound drugs [6,7].

Based on a history of safety and efficacy, buprenorphine has been the principal opioid to study LAI analgesia for veterinary medicine. Compared to morphine, buprenorphine is 25-fold more potent and 25-fold less toxic [8,9]. Four decades of clinical experience have confirmed its wide safety profile in veterinary medicine [10–12]. However, the translation of LAI analgesics from research to products for veterinary patients has lingered. Whether long-acting products and multiple bolus injections have similar efficacy has not been determined.

The efficacy of analgesia in animals has been evaluated using subjective behavioral observations and objective thermal and mechanical quantitative sensory testing (QST). A range of studies has demonstrated the analgesic biosimilarity of LAI opioids to multiple bolus doses of the legacy drug [13–18]. Investigators further evaluated analgesia by demonstrating the bioequivalence of buprenorphine in LAI formats and the generic drug given in single or multiple injections [19]. Drugs are confirmed to be bioequivalent if they enter the circulation and are accessible to the drug's receptor at the same rates [20].

Regulatory drug applications for generic and biosimilar veterinary analgesics first require data for drug safety. Test programs can exceed the safety protocols used to approve the legacy drug. The second requirement is bioavailability data, to provide evidence that a new product provides drug concentrations equivalent to the legacy drug. The kinetics of the drug in circulation are key determinants of its bioavailability. Bioavailability data can substitute for new rounds of efficacy trials, thereby decreasing both the time to approval as well as reducing the costs required for new studies [19].

This report reviews the pharmacokinetic (PK) data from over a decade of trials of extended-release polymer and lipid buprenorphine carriers in 13 animal species. Variations in the PK values among species, strains, and sex have been reported. Nonetheless, the data demonstrate that blood concentrations of buprenorphine delivered by several LAI products are equivalent to concentrations of the generic opioid which are known to provide analgesia. The evidence reviewed here introduces reasonable avenues for accelerated approval of LAI veterinary analgesic products based on combinations of generic buprenorphine with inert polymer and lipid carrier systems.

## 2. Methods

We reviewed the pharmacokinetic data associated with long-acting buprenorphine carrier products to treat acute pain in laboratory and companion animals, and nonhuman primates. The National Library of Medicine PubMed index was the primary search mechanism for this review. Citations were collected from 1970 to 2022. The search combined the term buprenorphine with the key words: blood concentrations, pharmacokinetics, mice, rat, dog, cat, and efficacy. Secondary searches were conducted on reports cited in the original search but not exposed in the primary collection. The secondary searches added 9 additional species. The clinical trial database of the US Food and Drug Administration (FDA) was searched for sustained release analgesia and anesthesia.

## 3. Results

Table 1 summarizes the PK data from the studies using a polymer-based buprenorphine LAI: Buprenorphine SR (Bup SR). The product is also sold as Extended Release Buprenorphine (Bup ER). In all cases, the dose was SC. The product was obtained from several compounding pharmacies: Wildlife Pharmaceutical (Windsor, CO, USA), Veterinary Technologies (Laramie, WY, USA), ZooPharm (Fort Collins, CO, USA), and Wedgewood Pharmacy (Swedesboro, NJ, USA). The PK studies used doses from 0.05 to 1.8 mg of

buprenorphine/kg to provide a theoretical minimum concentration (TMC) of buprenorphine for analgesia (see Discussion).

**Table 1.** Polymer-bound Buprenorphine PK Reports *.

| Species | Strain | Sex | Dose | AUC | Cmax | Tmax | QST | Reference |
|---|---|---|---|---|---|---|---|---|
| | | | mg/kg | h * ng/mL | ng/mL | h | | |
| Mouse | CD1 | F | 0.6 | 322 | 14.5 | 4 | no | [21] |
| Mouse | C57BL/6J | M | 0.3 | 20.1 | 0.8 | 6 | no | [22] |
| Mouse | C57BL/6J | M | 1.2 | 62.9 | 5.0 | 0.5 | no | [22] |
| Rat | Sprague Dawley | M | 1.2 | nd | 2.8 | 4 | yes | [23] |
| Rat | Norvegicus | M | 1.2 | nd | 1.2 | 24 | yes | [24] |
| Guinea Pig | Cavia porcelles | F | 0.3 | 56 | 1.3 | 1 | yes | [25] |
| Guinea Pig | Cavia porcelles | F | 0.15 | 50 | 2.0 | 1 | no | [26] |
| Guinea Pig | Cavia porcelles | F | 0.3 | 127 | 6.9 | 1 | no | [26] |
| Guinea Pig | Cavia porcelles | F | 0.6 | 1257 | 71.3 | 1 | no | [26] |
| Guinea Pig | Cavia porcelles | M | 0.15 | 32 | 2.3 | 1 | no | [26] |
| Guinea Pig | Cavia porcelles | M | 0.3 | 213 | 11.5 | 1 | no | [26] |
| Guinea Pig | Cavia porcelles | M | 0.6 | 1198 | 64.3 | 1 | no | [26] |
| Rabbit | New Zealand White | M | 0.15 | 22 | 0.6 | 39 | no | [27] |
| Prairie dog | Cynomys ludovicianus | MF | 0.9 | 624 | 191 ** | 8 | no | [28] |
| Prairie dog | Cynomys ludovicianus | MF | 1.2 | 863 | 17 | 24 | no | [28] |
| Dog | Beagle | F | 0.2 | 188.9 | 5.6 | 13.8 | no | [29] |
| Kestrel | Falco sparverius | MF | 1.8 | 665 | 69 | 0.25 | no | [30] |
| Seal | Mirounga angustirostris | MF | 1.2 | 93.8 | 1.2 | 12 | no | [31] |
| Göttingen minipigs | Sus scrofa domestica | F | 0.18 | 221.6 | 2.9 | 22.2 | no | [32] |
| Alpaca | | MF | 0.12 | 4.5 | 2 | 0.6 | no | [33] |
| Sheep | Suffolk | MF | 0.27 | 36.7 | 0.80 | 48 ** | yes | [34] |
| Sheep | Dorset & Suffolk | FF | 0.1 | nd | 0.1 | 48 | no | [35] |
| Sheep | Dorset & Suffolk | FF | 0.05 | nd | 0.1 | 72 | no | [35] |
| Marmoset | Callithrix jacchus | MF | 0.2 | 98.6 | 2.8 | 4 | no | [36] |
| Macaque | Mulata & fascicularis | MM | 0.2 | 177 | 15.3 | 9.3 | no | [37] |

* $T_{max}$, time of $C_{max}$; $C_{max}$, maximal concentration measured; AUC, Area under the concentration time curve from time 0 to last time point at which the concentration was measured or estimated. ** mean values.

Support for the hypothesis that the doses tested provided analgesia was confirmed by QSTs in rats [23,24] and a sheep study [35]. A guinea pig trial reported the duration of the mechanical QST-confirmed analgesia peaked at 12 h, a duration approximately 6 h longer than the duration of the analgesia obtained from the buprenorphine HCl injections (0.05 mg/kg; 0.3 mg/mL, Ricket Benckiser Healthcare, London, UK) twice daily for 60 h [26].

In the remaining studies reported in Table 1, pharmacokinetic support for the analgesic efficacy of the dose tested was extrapolated from the TMC literature values linking buprenorphine blood concentrations to analgesia in laboratory animals and humans. The mice, rat, guinea pig, and rabbit studies used doses from 0.15 to 1.2 mg/kg. Doses of 0.2–1.8 mg/kg were used in the rabbit, prairie dog, dog, kestrel, seal, minipig, alpaca, sheep, and non-human primate studies. An exception was a study with Dorset and Suffolk sheep, which used 0.05–0.1 mg/kg doses. In these sheep studies, the average plasma buprenorphine concentration was above 0.1 ng/mL at 48 h, up to 192 h post-injection for the 0.1 mg/kg dose group, and above 0.1 ng/mL at 48 h, up to 72 h post-injection for the lower-dose group. The authors reported that the effective analgesic plasma threshold still

needs to be determined in sheep [35]. A second exception was found in the alpaca studies. Plasma concentrations were detectable in only two of six alpacas after SC administration of Bup SR [33].

The area under the concentration versus time curve (AUC) describes the extent of the drug's bioavailability. AUC is a key parameter for comparing the bioequivalence of different drug formulations [38]. The values in Table 1 generally are proportional to the dose. For example, the doses of 0.15, 0.3, and 0.6 mg/kg in guinea pigs produced AUC values of 32-, 213-, and 1198-h*ng/mL, respectively [26]. Large differences in the relation between dose and AUC values are seen in seals [31] and kestrels [30]. Yet, the AUC calculations depend on the analytical sensitivity of the buprenorphine assay and if the concentrations of buprenorphine at the initial and final time points were measured or estimated. These interlaboratory analytical variables may account for the difficulty in comparing the values recorded in the different species, or within the same species studied in different laboratories.

The $C_{max}$ values in Table 1 show that the orthodox relation between dose and $C_{max}$ changes in LAI products. $C_{max}$ will be proportional to the dose in the studies with drugs injected intravenously. Rather, the bioavailability of LAI drugs depends on the release kinetics of the drug from the carrier. The data demonstrate that in the majority of studies the doses of the buprenorphine polymer product provided a sustained blood concentration of buprenorphine that has been associated with analgesia in TMC research: approximately 0.5–1 ng/mL. The range also suggests that the release kinetics of the polymer-bound product may vary with species, or that intra-experimental applications affect the release kinetics. A mean value of 191 ng/mL for four prairie dogs dosed with 0.9 mg/kg of Bup SR is notable. It results from averaging four buprenorphine plasma concentrations: 8, 12, 13, and 730 ng/mL. In the same study, four prairie dogs dosed with 1.2 mg/kg of Bup Sr generate a $C_{max}$ of 17 ng/mL [28].

Similar to the $C_{max}$ values, there was little relation between $T_{max}$ and the LAI dose (Table 1). The values ranged from 0.15 min in the kestrel [30] to 2–3 days in sheep [34,35]. When taken together with the $C_{max}$ data, the values in mice [21,22] and rats [24,25] given doses of 0.6–1.2 mg/kg demonstrate the TMC was achieved by 6 h. The studies in guinea pigs demonstrated that the TMC value was achieved within one hour of the dose [25,26]. The TMC values were likely achieved in 4–6 h in studies with the prairie dog [28], dog [29], kestrel [30], alpaca [33], and nonhuman primates [36,37].

Table 2 summarizes the PK data from the studies using a SC dose of lipid-based buprenorphine LAI: ER Buprenorphine (Ethiqa®, Fidelis Pharmaceuticals North Brunswick, NJ, USA). Compared to polymer-bound buprenorphine, there is significantly less published PK data with the lipid-bound LAI. Safety and efficacy data are available for mouse, rat, guinea pig, and dog. The studies used doses from 0.2 to 3.25 mg of buprenorphine/kg to provide a TMC of buprenorphine for analgesia.

**Table 2.** Lipid-bound Buprenorphine PK Reports.

| Species | Strain | Sex | Dose | AUC | Cmax | Tmax | QST | Reference |
|---------|--------|-----|------|-----|------|------|-----|-----------|
| | | | mg/kg | h * ng/mL | ng/mL | h | | |
| Mouse | BALB/c | MF | 3.25 | nd | 16.3 | 6 | no | [39] |
| Mouse | C57BL/6J | M | 3.25 | nd | 11.9 | 24 | yes | [40] |
| Mouse | C57BL/6J | M | 6.5 | nd | 19.4 | 24 | yes | [40] |
| Rat | Fischer F344/NTac | M | 0.65 | 154 | 3.4 | 24 | yes | [41] |
| Rat | Fischer F344/NTac | F | 0.65 | 100 | 1.8 | 24 | yes | [41] |
| Rat | Fischer F344/NTac | M | 1.3 | 459 | 6.6 | 48 | yes | [41] |
| Rat | Fischer F344/NTac | F | 1.3 | 517 | 9.6 | 24 | yes | [41] |

**Table 2.** *Cont.*

| Species | Strain | Sex | Dose | AUC | Cmax | Tmax | QST | Reference |
|---|---|---|---|---|---|---|---|---|
| Rat | Sprague-Dawley | M | 0.65 | 126 | 1.6 | 24 | yes | [42] |
| Rat | Sprague-Dawley | F | 0.65 | 86 | 1.2 | 6 | yes | [42] |
| Rat | Sprague-Dawley | M | 1.3 | 250 | 2.7 | 24 | yes | [42] |
| Rat | Sprague-Dawley | F | 1.3 | 175 | 1.8 | 24 | yes | [42] |
| Guinea Pig | Cavia porcelles | F | 0.48 | nd | 48 | 48 | yes | [43] |
| Dog | Beagle | MF | 0.2 | 224 | 5 | 8 | yes | [44] |

The lipid-bound LAI dose required to sustain the TMC of analgesia was greater in mice and lower in rats compared to the polymer-bound product. Shildhaus et al. demonstrated by thermal QST that chronic blood concentrations of 0.8–2 ng/mL for 24–72 h provided significant analgesia in male and female Swiss and BALB/c mice dosed with 3.1 mg/kg of lipid-bound buprenorphine [45]. As shown in Table 2, a dose of 3.25 mg/kg in mice provided buprenorphine blood concentrations greater than 1 ng/mL for 2–3 days [23,24]. A dose of 0.65 mg/kg provided 2–3 days of QST confirmed analgesia in Fischer and Sprague-Dawley rats [41,42]. Buprenorphine plasma concentrations exceeded 0.9 ng/mL from 8 to 96 h after injection of 0.48 mg/kg in guinea pigs [43]. A dose of 0.2 mg/kg provided a similar concentration profile for 80 h in dogs [44].

Values for the AUC and $C_{max}$ in Table 2 are proportional to the dose in the two rat studies available for comparison. The $T_{max}$ values range from 6 to 48 h. In every case, the blood concentrations of buprenorphine were greater than 1 ng/mL at $T_{max}$. This data demonstrates that the lipid-bound LAI provides increasing amounts of analgesia within hours of the dose.

## 4. Discussion

The TMC blood concentrations of buprenorphine providing analgesia in humans were estimated from the data in the first reported clinical trials determining whether buprenorphine was a safe substitute for morphine [46]. Whether given intravenous (IV) or intramuscular (IM) injections, the patients reported adequate to good relief for acute pain at 2 and 4 h. Buprenorphine blood concentrations reached approximately 5 ng/mL about 1 h after injection and fell to less than 0.5 ng/mL at 6–8 h after the initial dose [47,48]. These results were supported by the laboratory studies in rats using single injections and IV infusions of buprenorphine [49,50]. Nonetheless, the relationship between buprenorphine blood concentrations and sustained analgesia appeared uncertain into the mid-1980s for several reasons. One was the subjective nature of patient self-reports, especially in patients treated with buprenorphine for chronic pain [51]. A second reason concerned the chromatographic and radio immune assays used to detect buprenorphine in biological samples. Buprenorphine is rapidly converted to norbuprenorphine, its principal metabolite. Both are converted to glucuronides, which also have analgesic potential [52]. The third reason was the lack of biomarkers for analgesia in humans and animals. A fourth and perhaps major reason was the gradual unfolding of the understanding of buprenorphine's reaction with the morphine opiate receptor (MOR) [53,54] and the mechanisms of its tissue binding [55–57]. Studies in humans using buprenorphine for acute and chronic analgesia have confirmed that sustained buprenorphine blood concentrations from approximately 0.5 to 2 ng/mL are required to saturate the MOR. Studies have demonstrated virtually identical MOR receptor-binding properties in animals and humans [58]. Buprenorphine's receptor-binding characteristics provide both a necessary and sufficient explanation for the drug's analgesic and opioid use deterrence properties [19,59].

Lipid and polymer-bound buprenorphine products are logical choices for long-acting analgesic therapy. The latter has been used extensively for drug delivery to the brain [60]. The data in Tables 1 and 2 demonstrate that the first generation of long-acting analgesics for

laboratory animals that has provided clinically relevant levels of analgesia. Their efficacy has been confirmed by QSTs, which are the only objective measure for analgesia in animals. Numerous pain scales based on subjective behavioral observations have been published. None has been independently validated [61]. The discrepancy has been reported by Barletta et al. [62], and confirmed by investigations demonstrating the placebo effects in studies involving observations of animal behavior [63].

Blood concentrations of the opioid delivered by the implants are equivalent to concentrations of the generic opioid which are known to afford analgesia. Nonetheless, questions have arisen about the product reproducibility and safety. This is not surprising because real-world drug use frequently encounters parameters outside single-center trials conducted with a relatively small number of single-strain animals.

Because compounded drugs may not undergo consistent testing for identity, quality, strength, purity, and stability, the results of research described in reports using compounded products may not be reproducible. Different compounding processes may have played a role in the heterogeneity observed in the PK data in Table 1. Healey et al. reported that formulation changes improved Bup SR safety and analgesia properties in mice although the nature of the change was not specified [64]. Skin lesions have been linked to Bup SR in many species tested in Table 1. Haertel et al. calculated a 3% incidence rate in a study of 1559 injections in macaques [65]. Page et al. found 37 of 37 female athymic nude rats treated with Bup SR had nodules in the subcutis over the shoulders [66]. The nodules were identified as small, cystic structures (diameter, approximately 0.25 cm). The authors hypothesized a lack of T-cells prevented dissolution of the polymer particles.

Since an initial evaluation of Bup SR in rats [15], the reviews have shown the traditional method for manufacturing microspheres can produce heterogeneity in particle size and challenges for sterility [67]. Schreiner et al. designed a new microsphere formulation method to allow for a fast onset of action of buprenorphine and a duration of the analgesic effect of at least two days in laboratory mice. Drug release was characterized by an initial burst of 30% followed by complete release over seven days [68]. New methods have been described for producing buprenorphine polymers that decrease the initial burst release kinetics by half. Tests in rabbit demonstrated a 4.2 mg/kg dose yielded an AUC of 1719, a value about 100-fold greater than the value seen in Table 1 with the Bup SR product given at a 0.15 mg/kg dose [69]. Whether new manufacturing techniques can reduce the incidence of skin lesions that have been associated with SC implants is to be determined.

The results reviewed here summarize a remarkable body of studies conducted over a decade aimed at improving analgesic options for pain management in veterinary patients. Lipid and polymeric options for the design of LAI analgesics appear reasonable and eligible for accelerated approval regulatory pathways.

## 5. Addendum

Since this review was prepared, additional studies have been published confirming the utility of LAI buprenorphine analgesics for animal medicine and the relative absence of skin lesions at the injection site with a lipid-based drug carrier. PK studies in Nu/Nu and C57Bl/6J mice [70–72] and neonatal Sprague Dawley rats [73] confirmed and extended the evidence supporting LAI buprenorphine products in laboratory animal research.

**Author Contributions:** M.G. conceptualized and wrote the manuscript. B.M.T. edited the manuscript. All authors have read and agreed to the published version of the manuscript.

**Funding:** Funding for this research was supplied by The Maryland Biotechnology Center Biotechnology Development Awards and Maryland Industrial Partnerships (MIPS).

**Institutional Review Board Statement:** Not applicable.

**Informed Consent Statement:** Not applicable.

**Data Availability Statement:** Not applicable.

**Conflicts of Interest:** B.T. has no conflicts to report. M.G. holds significant financial interests in Peabody Pharmaceuticals and Animalgesics Laboratories, which are developing buprenorphine analgesics for human and veterinary medicine, respectively.

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
