# Peer review of "Long-Acting Opioid Analgesics for Acute Pain: Pharmacokinetic Evidence Reviewed"

_vetsci, doi:10.3390/vetsci10060372_

Round 1

Reviewer 1 Report

The review study is very interesting and provides useful information that must also have effective clinical feedback for the veterinary surgeon. Therefore, it is necessary to enrich it with studies that highlight ways of recognizing acute pain for the possible but still essential administration of rescue analgesia. In this regard, I suggest to the Authors to cite the acute pain assessment scales used in the various species, considered in the review and to cite and discuss the following publications which clearly highlight the method for establishing the cut of point for the administration of analgesia rescue in acute pain in some species.

Giovanna L Costa, Simona Di Pietro, Claudia Interlandi*, Fabio Leonardi, Daniele Macrì, Vincenzo Ferrantelli, Francesco Macrì. (2023) Effect on physiological parameters and anaesthetic dose requirement of isoflurane when tramadol given as a continuous rate infusion vs a single intravenous bolus injection during ovariohysterectomy in dogs. PLoS ONE 18(2 February), e0281602

2022 Claudia Interlandi, Simona Di Pietro, Giovanna L. Costa, Filippo Spadola, Nicola M. Iannelli, Daniele Macrì, Vincenzo Ferrantelli, Francesco Macrì. (2022). Effects of Cisatracurium in Sevoflurane and Propofol Requirements in Dog-Undergoing-Mastectomy Surgery. ANIMALS, p. 1-10, ISSN: 2076-2615, doi: 10.3390/ani12223134

Spadola, Filippo, Neve, Veronica Cristina, Interlandi, Claudia Dina, Spadaro, Andrea, Macrì, Francesco, Iannelli, Nicola Maria, Costa, Giovanna Lucrezia (2022). Hernioplasty with Peritoneal Flap for the Surgical Treatment of Umbilical Hernia in Swine. ANIMALS, vol. 12, p. 3240-3251, ISSN: 2076-2615, doi: 10.3390/ani12233240

Interlandi C., Leonardi F., Spadola F., Costa G. L. (2021). Evaluation Of The Paw Withdrawal Latency For The Comparison Between Tramadol And Butorphanol Administered Locally, In The Plantar Surface Of Rat, Preliminary Study. Plos One, P. 2-8, ISSN: 1932-6203, Doi: 10.1371

Leonardi F., Properzi R., Rosa J., Boschi P., Paviolo S., Costa G. L., Bendinelli C. (2020). Combined Laparoscopic Ovariectomy And Laparoscopic-Assisted Gastropexy Versus Combined Laparoscopic Ovariectomy And Total Laparoscopic Gastropexy: A Comparison Of Surgical Time, Complications And Postoperative Pain In Dogs. Veterinary Medicine And Science, p. 321-329, ISSN: 2053-1095

Costa G., Nastasi B., Spadola F., Leonardi F., Interlandi C. (2019). Effect Of Levobupivacaine, Administered Intraperitoneally, On Physiological Variables And On Intrasurgery And Postsurgery Pain In Dogs Undergoing Ovariohysterectomy. Journal Of Veterinary Behavior, vol. 30, p. 33-36, ISSN: 1558-7878, doi: 10.1016/j.jveb.2018.11.003

I also suggest highlighting the difficulty in recognizing pain in animals.

Author Response

Reviewer 1 [Questions/comments in bold italics]

Comments and Suggestions for Authors

The review study is very interesting and provides useful information that must also have effective clinical feedback for the veterinary surgeon. Therefore, it is necessary to enrich it with studies that highlight ways of recognizing acute pain for the possible but still essential administration of rescue analgesia. In this regard, I suggest to the Authors to cite the acute pain assessment scales used in the various species, considered in the review and to cite and discuss the following publications which clearly highlight the method for establishing the cut of point for the administration of analgesia rescue in acute pain in some species.

I also suggest highlighting the difficulty in recognizing pain in animals.

Response:

Reviewer 1 finishes and starts with contradiction.  The Reviewer suggests highlighting the difficulty in recognizing pain in animals.  And the review suggests we should “enrich” [our report] with studies that highlight ways of recognizing acute pain for the possible but still essential administration of rescue analgesia. 

                It has been known for 30 years that animals have an instinct to hide pain.  We also know there are no biomarkers of pain, chronic or acute.  There are few radiographic, biochemical, or physiological tests to estimate levels of pain in animals or humans.  Therefore, we do not agree that this report can be a mechanism to attempt to provide veterinary surgeons with clinical feedback regarding pain assessments.

Reviewer 1 may not be aware of over 10 years of international efforts to harmonize preclinical and clinical studies.  If a procedure is likely to cause pain in humans, the surgeon must assume it will cause pain in animals and provide appropriate analgesia.  For more than 20 years this has been an operational requirement of government agencies and NGO’s sponsoring animal research in the US, UK, and EU.  The concept of rescue analgesia becomes recessive with humane standards for animal analgesia. 

Reviewer 1 suggests we cite the acute pain assessment scales used in various species.  The Reviewer appears unaware that although dozens of acute pain assessment scales have been published, all depend on subjective behavioral assessments.  No assessment has been independently validated outside the laboratory claiming invention of the scale.  In 2018, an international US National Institute of Health conference on animal pain models concluded that no pain scale has the sensitivity or specificity to meet regulatory standards for drug or device development.   

We are grateful for the suggestion from Reviewer 1 to cite six reports, many from highly regarded Veterinary Institutes in Messina and Parma. The 1st by Costa et al., the 2nd from Interlandi et al., and the 3rd from Spadola et al., evaluate surgical and anesthetic protocols.  They are important but outside our evaluation of the pharmacokinetics of long-acting subcutaneous opiate implants. 

                Report 4 from Interlandi et al., compares the analgesic efficacy of two short-acting analgesics administered locally in the rat paw: tramadol and butorphanol.  We did not cite this report because we reviewed long-acting buprenorphine analgesia.  Tramadol and butorphanol must be given at 6–8-hour intervals.  Both drugs have lower margins of analgesic efficacy compared to buprenorphine.   Both drugs are associated with more adverse effects.  Efforts to secure and inject short term analgesia in surgically treated animals at 8-hour intervals over a 2–3-day period stress the animal and expose the animal to iatrogenic injuries.   

The 5th report from Leonardi et al., compares combined laparoscopic ovariectomy and laparoscopic-assisted incisional gastropexy with combined laparoscopic OIE and total laparoscopic gastropexy for surgical time, incidence of complications, and postoperative pain.  The authors cite several opinions that pain mainly occurs over the first 24 hr after surgery using minimally invasive techniques.  Pain was assessed for 24 hrs using the Glasgow short form with its visual observation scale.  Curiously, observers were not blinded to the treatment group.  The behavioral pain scale tool found no difference in the treatment groups.   Again, we commend the authors for their surgical research.  Yet the report is outside the scope of our study.    

The 6th report by Costa et al., evaluated “the effects of levobupivacaine combined with cisatracurium on akinesia and mydriasis when administered by peribulbar injection.”  Although the anesthetic report is yet another demonstration of the group’s surgical prominence, it too is beyond the scope of our analgesia review. 

mg, 05/11/23

Reviewer 2 Report

This is a good review.

I do have a couple of questions.

In line 82 Bup SR, is this sustained release, is helpful to spell out the abbreviations the first time they are used.

Why is the QST system used instead of a recognized pain scale for analgesic scoring? Could you give a bit more explaination on why the QST if preferred.

Line 89 how were the injections given, SC? In the studies cited please list how the bupreorphine was injected, SC. IV, or IM.

Line 137 Bup ER is this extended release, again spell out the abbreviations the first time they are used. 

What it the difference between ER and SR, the carrier?

Does the lipid -bound bupreorphine deliver higher, longer duration analgesia than the polymer-bound buprenorphine? Or are both equally effective? Is there less side-effects with one versus the other, less skin reactions with the lipid-bound buprenorphine? Do you recommend one over the other?

I didn't see this reference cited: The pharmacokinetics and analgesic effects of extended-release buprenorphine administered subcutaneously in healthy dogs. Barletta M, Ostenkamp SM, Taylor AC, Quandt J,  Lascelles BDX, Messenger KM.  J Vet Pharmacol Ther. 2018 Aug;41(4):502-512. 

Author Response

Reviewer 2 [Questions/comments in bold italics]

I do have a couple of questions.  In line 82 Bup SR, is this sustained release, is helpful to spell out the abbreviations the first time they are used. 

Response:           The Reviewer notes that we use the words Bup SR and Bup ER.  These are not abbreviations.  Rather, they are trade names of proprietary drug products.  Nonetheless, we have spelled out the abbreviations. 

Why is the QST system used instead of a recognized pain scale for analgesic scoring? Could you give a bit more explanation on why the QST if preferred.

Response:           The Reviewer asks about the use of quantitive sensory test (QST) testing rather than a recognized pain scale for analgesic scoring.  We have added the following language. “Quantitive sensory testing is the only objective measure for analgesia in animals.  Numerous pain scales based on subjective behavioral observations have been published.  Yet, none have been independently validated. [ number]  }This discrepancy has been reported by Barletta et al, [number] and confirmed by investigations demonstrating placebo effects in studies involving observations of animal behavior.[number]

Line 89 how were the injections given, SC? In the studies cited please list how the buprenorphine was injected, SC. IV, or IM.

Response:           All injections in this review were given SC.  We have added this information in Results section before describing the data in Table 1 and Table 2.

Line 137 Bup ER is this extended release, again spell out the abbreviations the first time they are used. 

What it the difference between ER and SR, the carrier?

Response:           Again, trade name abbreviations have been spelled out.  The Vendor, ZooPharm (Wildwood Pharmaceutical, Swedesboro NJ) changed the name of its polymer-based buprenorphine drug “Bup SR” to “Bup ER” in 2021.  Users are assured by the Vendor that Bup SR and Bup ER are the same product.  We are not in a position to verify the Vendors information. 

Does the lipid -bound buprenorphine deliver higher, longer duration analgesia than the polymer-bound buprenorphine? Or are both equally effective? Are there less side-effects with one versus the other, less skin reactions with the lipid-bound buprenorphine? Do you recommend one over the other?

Response:           The data in Table 1 and 2 directly speak to the Reviewer’s question.  The PK data demonstrate that in most cases the polymer and lipid bound implants provide comparable amount of drug.  There are fewer skin reactions with the lipid bound product.  

Regarding our preference, we declared a potential conflict of interest (COI) with the submission of this report. 

I didn't see this reference cited: The pharmacokinetics and analgesic effects of extended-release buprenorphine administered subcutaneously in healthy dogs. Barletta M, Ostenkamp SM, Taylor AC, Quandt J,  Lascelles BDX, Messenger KM. J Vet Pharmacol Ther. 2018 Aug;41(4):502-512. 

                 Barletta’s work on dogs is cited in Table 2:  Ref 45.

mg, 05/11/23

Reviewer 3 Report

The present review shows an interesting approach to the use of LAI opioids. These new presentations can improve availability and increase the possibility to control mild to moderate pain in animals. Therefore, I consider that this manuscript is suitable for publication. However, a weakness of this article is the lack of clarity about the type of review carried out and more discussion about the differences found in the selected species.

General comment: Please, revise the Instructions for Authors’ Guideline of the journal and amend the in-text citation style throughout the manuscript.

Line 1: Consider adding in the title that this is a review.

Line 18: In the keywords, the authors can add “pain”.

Lines 22-23: Add a reference for this statement. This article might help https://doi.org/10.5455/javar.2021.h529. Also, before starting with opioid use, consider briefly mentioning the importance of pain management and control with drugs such as opioids (e.g., to avoid the physiological consequences of pain perception in animals and how they can alter the postoperative period). These references could help Hernández-Avalos et al. (2019) Review of different methods used for clinical recognition and assessment of pain in dogs and cats Doi: https://doi.org/10.1080/23144599.2019.1680044.

Line 29: I suggest mentioning the half-life of each opioid because they differ among them. For example, fentanyl has a half-life of 2.5 h, morphine has 1.5 a 3 h, and buprenorphine around 5-6 h. Additionally, add references to the studies.

Lines 29-34: Apart from half-life, other important information such as bioavailability could also be mentioned.

Lines 35-41: I consider it important to describe in these lines the different types of LAI’s presentations that have been developed so far. This is relevant since their pharmacokinetics are highly influenced by the administration’s characteristics. This could serve the authors as an introductory text of why the authors decided to focus this review on polymer and lipid buprenorphine carriers.

Lines 39-40: This sentence could be changed to “It has been suggested that a single dose of subcutaneous LAI provided adequate analgesia to treat neuropathic pain for 2-3 days”.

Lines 65-70: Consider moving these lines to the discussion section. They provide more about the results than about the aim of the review.

Line 71, Methods: Please, include the search strategy the authors used to conduct the present review. Include exclusion and inclusion criteria of the articles (e.g., language, lack of certain information, different presentation, etc.). Also, in line 64 it is stated that “… carriers in 13 animal species”. However, in this section, only mice, rats, dogs, and cats are mentioned so the search is a little confusing.

Lines 158-163: I would suggest starting the discussion by highlighting the results of the present review. These lines seem to fit better in the introduction. For example, information about the doses, administration route, half-life, Cmax, and Tmax could be summarized according to the species that were used for the review. The discussion section is good, but I recommend focusing on three aspects: 1) Compare the advantages of LAI buprenorphine by extending its half-life and distribution volume; 2) Discuss its efficacy as an opioid to control mild to moderate pain (contrary to pure opioids that are recommended for severe pain); and 3) The lack of adverse effects when using LAI buprenorphine, in contrast to pure opioids such as morphine.

Lines 168-170: This aspect about the biotransformation of the drug is an interesting fact that could also be discussed, particularly because the authors reported different species, and is well known that not all animals can metabolize drugs in the same way and that affects the efficacy of the analgesic.

Line 217: Consider rephrasing this paragraph. Using the three aspects that I suggested could help to propose a solid conclusion that could enhance the relevance of the review.

Author Response

Reviewer 3 [Questions/comments in bold italics]

Comments and Suggestions for Authors

The present review shows an interesting approach to the use of LAI opioids. These new presentations can improve availability and increase the possibility to control mild to moderate pain in animals. Therefore, I consider that this manuscript is suitable for publication. However, a weakness of this article is the lack of clarity about the type of review carried out and more discussion about the differences found in the selected species.

Response:           We are grateful for the Reviewer’s suggestions and comments.  The topic of the review has been limited to LAI buprenorphine products that are presently available to a veterinary surgeon.  There are two exceptions: We did not review the use of transdermal buprenorphine patches because they are difficult to use and there are few reports describing their efficacy.  An additional concern is the abuse of these patches.  Owners may remove the patch from the injured animal to extract and divert the opiate for non-prescription use.  We did not review the use of the recently reported transdermal buprenorphine lotion for cats (TP Clark 2022).  There is too little information about the lotion’s efficacy and utility.   

                We agree that more discussion about the differences found in the species reported would be valuable.  We trust our review will facilitate it.  To be clear, we did not select the reported species.  We included every species reported.    

General comment: Please, revise the Instructions for Authors’ Guideline of the journal and amend the in-text citation style throughout the manuscript.

Response:           Done

 Line 1: Consider adding in the title that this is a review.

 Response:          Done

Line 18: In the keywords, the authors can add “pain”.

 Response:          We did not use the keyword “pain.”  Our review focuses on analgesia.    

Lines 22-23: Add a reference for this statement. This article might help https://doi.org/10.5455/javar.2021.h529. Also, before starting with opioid use, consider briefly mentioning the importance of pain management and control with drugs such as opioids (e.g., to avoid the physiological consequences of pain perception in animals and how they can alter the postoperative period). These references could help Hernández-Avalos et al. (2019) Review of different methods used for clinical recognition and assessment of pain in dogs and cats Doi: https://doi.org/10.1080/23144599.2019.1680044.

Response:           Done.  We have added the reference.  It further supports our response to Reviewer 1 that although dozens of acute pain assessment scales have been published, all depend on subjective behavioral assessments.  No assessment has been independently validated outside the laboratory claiming invention of the scale.  

Line 29: I suggest mentioning the half-life of each opioid because they differ among them. For example, fentanyl has a half-life of 2.5 h, morphine has 1.5 a 3 h, and buprenorphine around 5-6 h. Additionally, add references to the studies.

Response:           We thank the Reviewer for noting the discrepancy.  We removed the reference to fentanyl with its 0.5-1 hr half-life.

Lines 29-34: Apart from half-life, other important information such as bioavailability could also be mentioned.

 Response:          Drug regulatory agencies allow PK data to define bioavailability.  We have given this information.

Lines 35-41: I consider it important to describe in these lines the different types of LAI’s presentations that have been developed so far. This is relevant since their pharmacokinetics are highly influenced by the administration’s characteristics. This could serve the authors as an introductory text of why the authors decided to focus this review on polymer and lipid buprenorphine carriers.

Response:           Numerous LAI presentations have been described at research conferences and patent offices.  Several have been developed in our laboratory.  Yet only two have US Food and Drug Administration registration: the polymer and lipid-based product.  Therefore, we reviewed those two products. 

 Lines 39-40: This sentence could be changed to “It has been suggested that a single dose of subcutaneous LAI provided adequate analgesia to treat neuropathic pain for 2-3 days”.

Response:  The change is not warranted based on published results using both LAI products.  We are not aware that that suggestion has been published, or whether it discriminates nociceptive and neuropathic pain. 

Lines 65-70: Consider moving these lines to the discussion section. They provide more about the results than about the aim of the review.

Response:           Here and with additional comments, the Reviewer appreciates we are the first to review this topic.  Our goal is to provide a platform for further studies and emphasize the need for better analgesics for veterinary medicine.  Thus, we support the present organization.

 Line 71, Methods: Please, include the search strategy the authors used to conduct the present review. Include exclusion and inclusion criteria of the articles (e.g., language, lack of certain information, different presentation, etc.). Also, in line 64 it is stated that “… carriers in 13 animal species”. However, in this section, only mice, rats, dogs, and cats are mentioned so the search is a little confusing.

 Response:          The search strategy is described in Methods.  We changed the Section to cite the inclusion of 9 more species. 

Lines 158-163: I would suggest starting the discussion by highlighting the results of the present review. These lines seem to fit better in the introduction. For example, information about the doses, administration route, half-life, Cmax, and Tmax could be summarized according to the species that were used for the review. The discussion section is good, but I recommend focusing on three aspects: 1) Compare the advantages of LAI buprenorphine by extending its half-life and distribution volume; 2) Discuss its efficacy as an opioid to control mild to moderate pain (contrary to pure opioids that are recommended for severe pain); and 3) The lack of adverse effects when using LAI buprenorphine, in contrast to pure opioids such as morphine.

Response:           The suggestions for rewriting our Discussion focusing on three aspects recommended by the Reviewer are excellent but would redirect our focus.  The Reviewer appears to confuse the therapeutic properties of opiates and opioids.  Buprenorphine, a derivative of the opioid morphine has 25-30-fold more analgesic effect compared to morphine for the control of moderate to severe pain.   

Lines 168-170: This aspect about the biotransformation of the drug is an interesting fact that could also be discussed, particularly because the authors reported different species, and is well known that not all animals can metabolize drugs in the same way and that affects the efficacy of the analgesic.

Response:           We agree.  Drug metabolism could be discussed for the different species.  Our review provides potential reviewers with a logical starting point.  

Line 217: Consider rephrasing this paragraph. Using the three aspects that I suggested could help to propose a solid conclusion that could enhance the relevance of the review.

 Response:          The three suggested aspects indeed furnish a foundation for several reviews.  However, we prefer to maintain attention on the evidence reviewed here.  This focus introduces reasonable avenues for accelerated approval of LAI veterinary analgesic products based on combinations of generic buprenorphine with inert polymer and lipid carrier systems. 

mg, 05/11/23